# Proteostasis Perturbations and Their Roles in Causing Sterile Inflammation and Autoinflammatory Diseases

**DOI:** 10.3390/cells11091422

**Published:** 2022-04-22

**Authors:** Jonas Johannes Papendorf, Elke Krüger, Frédéric Ebstein

**Affiliations:** Institut für Medizinische Biochemie und Molekularbiologie, Universitätsmedizin Greifswald, Ferdinand-Sauerbruch-Straße, 17475 Greifswald, Germany; jonasjohannes.papendorf@med.uni-greifswald.de

**Keywords:** proteostasis, autoinflammation, ribosomopathies, proteinopathies, proteasomopathies

## Abstract

Proteostasis, a portmanteau of the words protein and homeostasis, refers to the ability of eukaryotic cells to maintain a stable proteome by acting on protein synthesis, quality control and/or degradation. Over the last two decades, an increasing number of disorders caused by proteostasis perturbations have been identified. Depending on their molecular etiology, such diseases may be classified into ribosomopathies, proteinopathies and proteasomopathies. Strikingly, most—if not all—of these syndromes exhibit an autoinflammatory component, implying a direct cause-and-effect relationship between proteostasis disruption and the initiation of innate immune responses. In this review, we provide a comprehensive overview of the molecular pathogenesis of these disorders and summarize current knowledge of the various mechanisms by which impaired proteostasis promotes autoinflammation. We particularly focus our discussion on the notion of how cells sense and integrate proteostasis perturbations as danger signals in the context of autoinflammatory diseases to provide insights into the complex and multiple facets of sterile inflammation.

## 1. Introduction

Inflammation describes a highly conserved generally beneficial reaction—if controlled—of the body in response to a variety of danger signals [1,2,3]. It is triggered by the innate immune system and characterized by the rapid mobilization of specialized cells and humoral factors which combat the insult with a relatively low specificity [4,5,6]. Noxious signals capable of initiating innate responses are traditionally divided into exogenous and endogenous insults in order to facilitate self and non-self-discrimination [7,8]. External stimuli are typically derived from infectious agents and include a wide range of microbial products referred to as pathogen-associated molecular patterns (PAMPs) which are recognized by pathogen-recognition receptors (PRR) such as Toll-like receptors (TLR), RIG-I-like receptors (RLR), NOD-like receptors (NLR), among others [9,10]. The binding of PAMP to PRR triggers multiple intracellular signaling cascades ultimately resulting in the activation of the NF-*κ*B and IRF3 transcription factors which in turn promote the expression of pro-inflammatory cytokines (i.e., TNF*α*, IL-1, IL-6) and type I and/or II interferons (IFN) [11,12]. Type I IFN is a large cytokine family consisting of 18 members with 14 IFN-*α* species and one single specimen each of IFN-*β*, IFN-*κ*, IFN-*ω* and IFN-ε which are produced by both immune and non-immune cells [13]. Following autocrine and/or paracrine binding to their receptor IFNAR1/2, type I IFN engage JAK/STAT signaling that results in the rapid transcription of a myriad of so-called IFN-stimulated genes (ISG) [14]. Prominent ISG include *PKR* and *OAS1* whose products confer the cell an antiviral state by arresting protein translation and promoting RNA degradation, respectively [14]. Internal stimuli also known as damage-associated molecular patterns (DAMPs) are widely regarded as self-molecules that accidently reach undedicated compartments following tissue injury [15,16]. Prime examples of DAMPs include extracellular and/or cytosolic DNA as well as extracellular ATP or histones, only to name a few [17,18,19,20]. Like PAMPs, DAMPs use various PRR to elicit innate immune responses which are characterized by the prompt production of NF-*κ*B- and IRF3-responsive genes [21,22]. As innate immunity triggered by DAMPs is pathogen-free, it is frequently categorized as sterile inflammation [23,24]. Importantly, sterile inflammation is not necessarily restricted to the unphysiological release of molecules from necrotic cells and/or damaged intracellular organelles. Disturbances in protein flux have also long been known to be immunostimulant and prominent hallmarks of a flurry of autoimmune diseases. It is indeed more than 60 years ago that extracellular protein aggregates initially called “rheumatoid factors” have been described in the plasma and synovial tissues of patients with rheumatoid arthritis (RA) [25]. It became rapidly evident that such aggregates were in fact large insoluble autoimmune complexes (IC) made up of self-reactive immunoglobulin (Ig) G [26]. It is their inefficient clearance by circulating proteases and/or phagocytes that results in their deposition in tissues and contributes to autoinflammation, notably through the stimulation of monocytes, macrophages and dendritic cells (DC) [27,28]. Herein, RA provided the first example of sterile inflammation triggered by loss of protein homeostasis. Meanwhile, over 40 autoinflammatory diseases have been reported to be associated with protein homeostasis (or more commonly referred to as “proteostasis”) perturbations.

## 2. Proteostasis Sensors and Their Roles in Triggering Sterile Inflammation

Imbalances of protein flux are not limited to the extracellular space but can also occur within the cell as a consequence of a disequilibrium between protein synthesis at ribosomes (i.e., translation) and degradation by the ubiquitin–proteasome system (UPS) and/or autophagy [29]. Regardless of localization, it is becoming increasingly clear that proteostasis perturbations behave as danger signals and have the potential to initiate sterile inflammation. Although our current knowledge on the mechanisms perceiving and integrating these signals are not fully understood, a growing body of evidence suggests a critical role of the endoplasmic reticulum (ER) in the surveillance of intracellular proteostasis. The mechanisms monitoring proteostasis outside the cell are less clear, albeit a couple of studies point to a potential involvement of PRR in this process, as discussed below.

### 2.1. The Unfolded Protein Response (UPR)

As shown in Figure 1, the ER-membrane resident proteins IRE1, ATF6 and PERK belong to the mechanisms by which the cell is capable of sensing intracellular proteostasis disruption. One estimates that approximately 30–40% of the newly synthetized proteins traffic through the endoplasmic reticulum (ER) [30]. It is further understood that up to 25% of these proteins fail to achieve final conformation and that such products are recognized by the protein quality control machinery in the ER and transported back to cytosol by the ER-associated degradation (ERAD) pathway for subsequent degradation by 26S proteasomes [31]. Given that retro-translocation is driven by degradation [32,33,34], proteasome dysfunction ultimately results in the accumulation of misfolded proteins within the ER lumen and subsequent activation of the IRE1, ATF6 and PERK receptors. This in turn engages the so-called unfolded protein response (UPR), a compensatory reaction which mostly aims to restore proteostasis by
Arresting global protein biosynthesis;Upregulating specific genes coding for chaperones and ERAD components [35,36,37].

While transient stimulations of the UPR are physiological reactions devoid of inflammation, excessive UPR activation is considered pathological and associated with persistent innate responses [35,38]. The mechanisms by which the UPR triggers sterile inflammation are diverse, complex, poorly understood and have been already discussed elsewhere [35,39]. Of particular interest is the PERK arm of the UPR which mediates the phosphorylation of eIF2*α*, an event that shuts down canonical translation of 5′ capped mRNA molecules and affects the steady-state level of short-lived proteins [40]. These proteins include notably IκB*α* whose increased turnover upon activation of the UPR facilitates the nuclear translocation of NF-*κ*B and the subsequent production of inflammatory mediators [41,42]. As discussed later, the UPR has been proposed as a major contributor to autoinflammation in various proteinopathies [35].

### 2.2. The Integrated Stress Response (ISR)

As proteasome-mediated protein degradation produces peptides which are further degraded into amino acids by peptidases [43], it is legitimate to assume that the intracellular pool of amino acids is regulated—at least partially—by the UPS. This hypothesis, which has been proven to be true in yeast [43], implies that proteostasis perturbations caused by impaired protein breakdown are followed by a parallel depletion of intracellular amino acids.

A major sensor of amino acid deficiency in the cell is the general control nonderepressible 2 (GCN2) kinase which undergoes activation by uncharged tRNA that accumulate upon amino acid starvation [44]. As shown in Figure 1, like PERK, GCN2 promotes the phosphorylation of eIF2*α* in a pathway referred to as the integrated stress response (ISR) [45] which, by definition, intersects with the UPR. As both the ISR and PERK branch of the UPR converge to eIF2*α*, these pathways share similar consequences regarding their proinflammatory properties in case of sustained stimulus. Another kinase of the ISR capable of phosphorylating eIF2*α* is protein kinase R (PKR) [46], which is primarily activated by double-stranded (ds) RNA in infected cells during viral replication [47,48]. Besides phosphorylating eIF2*α*, PKR also triggers a signaling cascade resulting in the nuclear translocation of the transcription factor IRF3 and subsequent transcription of type I IFN genes [49,50,51,52]. Interestingly, the PKR activation spectrum is not restricted to RNA viruses and can be expanded to a flurry of non-microbial stimuli including proteostasis disruptors such as tunicamycin, oxidative stress and heat shock [53,54,55,56,57,58]. A recent study by Davidson et al. described cytosolic IL-24 as a potent PKR activator following proteasome dysfunction [59]. In this process, misfolding of newly synthetized IL-24 within the ER lumen results in its retro-translocation back to the cytosol by ERAD where it accumulates and activates PKR to engage a type I IFN response. Supporting the notion that abnormal cytosolic accumulation of IL-24 is a danger signal alerting the immune system via PKR, cells devoid of IL-24 and/or PKR fail to exhibit a type I IFN signature in response to proteasome inhibition [59]. The work of Davidson et al. therefore unambiguously identifies the IL-24/PKR axis as a new surveillance pathway capable of initiating sterile inflammation in response to perturbed proteostasis. This notion is fully in line with the observation that PKR is found constitutively phosphorylated/activated in the CNS of patients suffering from neurodegenerative proteinopathies [60,61,62,63] and that PKR inhibition is able to delay neuroinflammation in rats [64].

### 2.3. The mTORC1 Signaling Complex

Apart from GCN2, major amino acid sensors include the cytosolic SESN2 and CASTOR proteins which sense leucine and arginine levels, respectively [65]. Binding of these amino acids to their respective receptors activate GATOR2, which in turn inhibits the mTORC1 complex inhibitor GATOR1 [66]. Subsequently, mTORC1 itself undergoes activation and engages several downstream signaling pathways inhibiting autophagy, while promoting protein translation, lipid biosynthesis and mitochondria biogenesis [67]. Based on the assumption that the UPS regulates amino acid homeostasis, one would expect that impaired proteostasis result in downregulation of mTORC1 signaling (Figure 1). Indeed, our recent investigation on cells isolated from patients with *PSMD12* haploinsufficiency confirmed that proteasome loss-of-function was associated with reduced activation of mTORC1 [68]. From the mTORC1 downstream pathways, lipid biosynthesis seems of particular interest regarding sterile inflammation. It was indeed recently suggested that a reduced cholesterol flux in the cell engages spontaneous type I IFN responses in a poorly understood process involving the cGAS/STING pathway [69]. Likewise, increased autophagy by reduced mTORC1 activation might be relevant with respect to sterile inflammation following proteotoxic stress. Elevated autophagy does not only imply increased lysosomal protein breakdown but also presupposes increased elimination of intracellular organelles including mitochondria, a process referred to as “mitophagy”. Supporting the notion that mTORC1 is a proteostasis sensor, increased mitophagy has been observed in patients with proteasomopathies [68]. Interestingly, sustained mitophagy implies a switch of the energetic metabolism from oxidative phosphorylation to glycolysis, the latter being associated with the accumulation of PEP, succinate, citrate and itaconate all exhibiting pro-inflammatory properties (i.e., M1 phenotype) by various and sometimes poorly understood mechanisms [70]. Whether mitophagy indeed participates in the initiation of sterile inflammation in response to impaired proteostasis is not known, but the observation that glycolysis is increased in microglia and astrocytes of patients in AD [70,71,72] supports this notion.

### 2.4. Stress-Induced Granulation

A cellular feature frequently shared by many neurodegenerative proteinopathies is the formation of so-called stress granules (SGs) in the CNS [73,74,75,76,77,78,79,80,81,82,83,84]. SGs are cytoplasmic liquid-like seemingly compartmentalized structures devoid of membranes mostly containing translation factors, RNA as well as RNA-binding proteins [85,86,87,88]. Importantly, SGs are not an exclusive feature of neurodegenerative diseases, and their formation may be induced in all cell types in response to various stress reactions driven by eIF2*α* phosphorylation, namely following activation of the UPR and/or ISR (Figure 1) [89]. Accordingly, SGs are induced in situations in which the proteostasis network is challenged such as viral infection and proteasome inhibition and [90,91,92]. Interestingly, while the role of SGs in the propagation of type I IFN responses during viral infection seems to be established [93,94,95], their contribution to inflammation following sterile proteostasis disruption is not known. Viral SGs are thought to amplify innate immune signaling by building a platform that facilitates the recruitment of the DNA/RNA sensors cGAS, PKR and RIG-1 [96,97,98]. It is unclear whether viral SGs are biochemically distinct from those observed in proteinopathies. In this regard, further investigations are warranted to answer the question whether SGs due to impaired proteostasis of non-viral origin predispose the cells to generate type I IFN responses.

### 2.5. The TCF11/Nrf1-NGLY1-DDI2 Axis

Another proteostasis sensor is the ER membrane-resident protein TCF11/Nrf1, also referred to as nuclear factor erythroid derived 2-related factor 1 (NFE2L1) and encoded by the *NFE2L1* gene [99,100]. TCF11/Nrf1 is a highly glycosylated short-lived protein which under normal conditions is rapidly targeted to proteasome-mediated degradation by ERAD [99,100] following its de-glycosylation by the enzyme N-glycanase 1 (NGLY1) [101]. Consequently, in case of proteasome dysfunction, TCF11/Nrf1 becomes stabilized and prone to enzymatic processing by the aspartyl protease protein DDI1 homolog 2 (DDI2) at the ER membrane [102]. As shown in Figure 1, following proteolytic cleavage, processed TCF11/Nrf1 translocates into the nucleus and binds to ARE promoters to induce the transcription of 20S and 19S proteasome subunit genes and a subset of ubiquitin factors. The TCF11/Nrf1-DDI2 pathway is widely viewed as a compensatory mechanism ensuring the replacement of defective 26S complexes by newly synthetized ones. Like all proteostasis sensors, TCF11/Nrf1 seems to have the potential to engage proinflammatory programs. This assumption is based on the observation that TCF11/Nrf1 also up-regulates mitophagy genes [103]. While the safe elimination of damaged mitochondria by autophagy prevents the accidental leakage of immunostimulatory mitochondrial (mt) DNA into the cytosol, the removal of the healthy ones due to excessive mitophagy might reprogram the cells towards glycolytic processes and their inherent proinflammatory consequences. On the other hand, TCF11/Nrf1 has been shown to mitigate lipid-mediated inflammation acting as a sensor for cholesterol [104]. Altogether, these studies highlight that our current understanding of TCF11/Nrf1 in inflammation remains incomplete and warrants further investigation.

### 2.6. Pathogen Recognition Receptors (PRR)

Unexpectedly, works aimed to unravel the pathogenesis of neurodegenerative proteinopathies have revealed that PRR are able to sense intra- and extracellular proteostasis perturbations. Indeed, normally specialized in the recognition of PAMP and/or DAMPs, PRR are also capable of perceiving abnormal protein aggregates. For instance, extracellular *α*-syn oligomers and fibrils produced during the course of Parkinson disease (PD) bind to TLR4 and TLR2 expressed on microglia [105,106,107,108,109,110]. In a very similar fashion, A*β* aggregates from Alzheimer disease (AD) are captured by TLR2, 4, 6 and 9 [111,112,113,114,115]. In ALS, the inflammasome is activated by cytosolic TDP-43 aggregates [116,117,118]. These observations suggest a role for PRR beyond sensing PAMP and/or DAMP in surveilling both intra- and extracellular proteostasis disturbances. It remains however unclear whether this perception is limited to certain types of proteins and/or aggregates.

## 3. Causes of Proteostasis Perturbations and Associated Autoinflammatory Syndromes

Depending on the source of perturbation, proteostasis imbalances carry two possible consequences, namely
Protein aggregation or;Protein depletion.

As illustrated in Figure 2, protein aggregation is frequently found in disorders such as proteinopathies or proteasomopathies following excessive protein misfolding or proteasome dysfunction, respectively. By contrast, intracellular protein depletion mostly occurs as a consequence of impaired translation in so-called ribosomopathies. The investigation of the molecular pathogenesis of such disorders have greatly improved our knowledge on how the innate immune respond reacts to disrupted proteostasis, as discussed below.

### 3.1. Proteostasis Perturbations Caused by Translation Deficiency

Defects in protein synthesis are typically due to loss-of-function mutations in genes encoding components of the translation machinery. Prominent vulnerable genes in this category include those coding for proteins of the 40S and 60S ribosomal subunits (RPS and RPL, respectively). Genomic alterations of the RPS and/or RPL genes generally result in ribosome assembly disruption which itself causes a wide range of disorders traditionally referred to as “ribosomopathies” [119].

At the molecular level, this group of syndromes is essentially characterized by reduced protein synthesis due to persistently depressed translation [120]. Accordingly, ribosomopathies primarily affect cells with high protein demand, particularly those of the hematopoietic lineage. Major ribosomopathies include the various forms of Diamond-Blackfan anemia (DBA), which are inherited monogenic autosomal dominant genetic disorders characterized by ineffective erythropoiesis and caused by mutations in either RPS or RPL genes [121]. Here, insufficient protein synthesis leads to protein depletion and cell death. Specifically, it is understood that the ribosomal subunits that fail to become incorporated into ribosomes initiate a so-called nucleolar stress response that results in cell cycle arrest and apoptosis [122,123]. Paradoxically, ribosomopathies also increase cancer risk overtime [124]. This seemingly inconsistency can be however easily explained by the fact that cell proliferation defects exert a strong pressure that results in the progressive selection of transformed cells that have acquired compensatory mutations in proto-oncogenes [124].

Most importantly, it was recently suggested that proteostasis disruption in DBA was associated with the generation of inflammatory gene signatures [125,126,127]. Autoinflammation could be confirmed in a DBA zebrafish model in which affected animals constitutively express type I IFN-stimulated genes (ISG) [128]. Further evidence for a functional cause-and-effect relationship between translation deficiency and sterile inflammation comes from the observation that mutations in RPL10 drive type I IFN responses in T cell leukemia [129]. The mechanisms by which ribosomopathies favor autoinflammation remain unclear, but it is conceivable that senescence due to prolonged cell cycle arrest might play a role in this process. Indeed, as shown in Figure 3, it was recently shown that senescence is intrinsically associated with genome instability and the subsequent leakage of DNA fragments into the cytosol which trigger type I IFN response by the cGAS/STING pathway [130,131]. Consistent with a potential role of cGAS in driving inflammatory responses upon ribosomal dysfunction, Wan et al. have shown that ribosome collision during translation results in the cytosolic accumulation of cGAS [132], thereby predisposing the cells to respond stronger to cytosolic, even background, DNA (Figure 3).

### 3.2. Proteostasis Perturbations Caused by Protein Misfolding

It is well established that protein biosynthesis (or mRNA translation) in eukaryotic cells is not an accurate process. Early works from Yewdell’s lab estimate that approximately 30% of newly synthetized proteins fail to reach their 3D structure because of damage, premature termination and/or incorporation of wrong amino acids [133,134]. As these improperly folded proteins (or DRiPs) are non-functional and, in most cases, prone to aggregation, they are rapidly sampled and sorted out by various protein quality control (PQC) systems. One major co-translational PQC mechanism relies on the action of the molecular chaperones Hsp70 and Hsp90 that are capable of sensing hydrophobicity exposed by all misfolded nascent proteins [135]. Both Hsp70 and Hsp90 recruit E3 ubiquitin ligases such as the carboxy-terminus of the Hsc70 interacting protein (CHIP), leucine-rich repeat and sterile alpha motif-containing 1 (LRSAM1) and/or E6-associated protein (E6-AP) which themselves mediate the ubiquitination of the presented protein substrates and target them for subsequent degradation by the 26S proteasome [136,137].

Besides aging, major causes for excessive misfolding include alterations in genes coding for
(i)UPS components or;(ii)Highly translated host (or viral) proteins.

These events may result in the accumulation of insoluble protein aggregates, a typical feature of a heterogenous group of disorders known as proteinopathies [138,139]. As mentioned above, proteostasis disruption in proteinopathies is not necessarily due to UPS dysfunction. Indeed, virtually every protein variant with a particularly high synthesis rate may increase protein supply and/or misfolding, thereby surpassing UPS capacity. Prime examples of protein mutants predisposed to misfolding and aggregating include the HLA-B27 allele as well as various variants of the cystic fibrosis transmembrane conductance regulator (CFTR), transthyretin (TTR), *α*-1 antitrypsin (AAT), amyloid protein precursor (APP), *α*-synuclein (SNCA), huntingtin (HTT) and transactive response RNA/DNA-binding protein (TDP-43), just to name a few. With the exception of HLA-B27, CFTR and AAT, all these proteinopathies primarily affect CNS and promote typical neurodegenerative phenotypes. The reasons for this tissue selectivity remain obscure and intriguing in view of the wide distribution of these proteins throughout the body. An undebatable issue in this field, however, is that neurodegeneration has a sterile inflammatory component—commonly referred to as neuroinflammation—predominantly triggered by glia cells (i.e., microglia, astrocytes and/or oligodendrocytes). Likewise, non-neurodegenerative proteinopathies are strongly associated with autoinflammation. These observations unambiguously point to a cause-and-effect relationship between protein aggregation and innate immunity, as discussed below.

#### 3.2.1. Non-Neurodegenerative Proteinopathies

These disorders are essentially caused by misfolding and subsequent aggregation of protein variants outside the CNS. These typically include ankylosing spondylitis (AS), cystic fibrosis (CF), alpha-1-antitrypsin deficiency (AATD) and hereditary transthyretin amyloidosis (ATTR) which are all associated with inflammatory reactions. As a large fraction of these mutant proteins (i.e., HLA-27, CFTR and AATD in AS, CF and AATD, respectively) typically misfold within the ER lumen, persistent activation of the UPR is thought to contribute to the disease inflammatory phenotype [140,141,142,143,144,145,146,147,148]. Nevertheless, additional pathways may synergize with the UPR to trigger autoinflammation in these patients.

For instance, ATTR is an autosomal dominant proteinopathy with neuronal and cardiac manifestations which provides an excellent example of extracellular proteostasis perturbation. It is characterized by the extracellular deposition of transthyretin (TTR)-derived amyloid fibrils, particularly in the peripheral nervous system (PNS) [149]. The pathologic basis of ATTR are single amino acid variations in the TTR protein, among which the Val30Met and Val142Ile substitutions are the most frequent ones [150]. The TTR protein is produced by the choroid plexus and liver and found in the cerebrospinal fluid (CSF) as well as in the bloodstream where it carries thyroxin and retinol [35,151]. TTR misfolding mutations typically result in the formation of extracellular precipitates resembling β-pleated sheet structures named amyloid fibrils which cause tissue dysfunction [152]. Although patients with ATTR exhibit no evidence of systemic inflammation, they present with elevated circulating inflammatory markers [153,154]. Recent evidence suggests that activation of the innate immune system in ATTR occurs extracellularly as well with TTR aggregates capable of activating immune cells, notably neutrophils and microglia to produce proinflammatory cytokines [155,156]. The receptors involved in this process remain however ill-defined.

#### 3.2.2. Neurodegenerative Proteinopathies

This group of proteinopathies is by far the largest and includes prominent disorders such as Alzheimer’s disease (AD), Parkinson’s disease (PD), Huntington’s disease (HD), spinocerebellar ataxias and polyglutamine (polyQ) diseases, among others. Although these disorders are clinically and phenotypically distinct, their molecular etiology is identical and based on the uncontrolled accumulation of protein variants mostly in the CNS. As alluded to earlier, one major mechanism by which proteostasis perturbations trigger neuroinflammation in these diseases involves the release of protein aggregates into the extracellular space and their subsequent binding to PRR such as TLR and/or TREM2 on microglia [105,106,107,108,109,110,111,112,113,114,115]. However, recent research on the molecular pathogenesis of amyotrophic lateral sclerosis (ALS) has expanded the spectrum of the molecular pathways initiating sterile inflammation in response to proteostasis disruption.

ALS is a neurodegenerative proteinopathy characterized by a progressive loss of motor neurons [157,158]. While most cases of ALS are sporadic, a dozen of predisposition genes have been identified over the past two decades [159,160]. These include notably SOD1 and TARDBP which code for the superoxide dismutase 1 (SOD1) and transactive response DNA binding protein 43 (TDP-43), respectively [161,162]. Both proteins tend to misfold and aggregate in neurons upon specific point mutations. In particular, abnormal accumulation and/or localization of TDP-43 accounts for 90% of ALS cases [163]. TDP-43 is a ribonucleoprotein highly expressed in the CNS with functions in RNA processing [164,165] and its deposition is favored by aging, variations in the TDP-43 gene itself and/or in the C9orf72, GRN or TBK1 genes. [166,167,168,169,170,171]. Not surprisingly, ALS is associated with constitutive activation of microglia and ongoing neuroinflammation [172,173]. The mechanisms involved in the inflammatory component of ALS are diverse and seem to depend on genetic lesion. For instance, TDP-43 or its variants, as cytosolic proteins not trafficking in the ER, do not generate ER stress and/or induce the UPR [174]. By contrast, SOD1 mutant species, although devoid of signal sequence, have been shown to impair ERAD and activate the UPR [175], which itself conceivably contributes to autoinflammation (Figure 4). In addition, SOD1 aggregates are released to activate microglia via binding to TLR2, TLR4 and/or CD14 (Figure 4) [176]. The neuroinflammatory phenotype of individuals with ALS is also thought to be a direct consequence of aggregation of TDP-43 or its mutant in the mitochondria [177,178]. Indeed, a recent study showed that mislocalization of TDP-43 variants in mitochondria results in mitochondrial DNA leakage into the cytosol, subsequent activation of the cGAS/STING innate pathway and IFN signaling in ALS. (Figure 4) [118]. In addition, cytosolic TDP-43 aggregates may promote innate immune responses by activating the inflammasome (Figure 4) [116,117]. Whether these processes occur simultaneously is unclear and their respective contributions to neuroinflammation may be also difficult to assess. In any case, these works reveal that the ability of the cells to react to impaired proteostasis due to protein misfolding is broad and probably even wider than initially assumed.

### 3.3. Proteostasis Perturbations Caused by Impaired Protein Degradation

As alluded to earlier, the breakdown of intracellular proteins in eukaryotic cells is ensured by two major conserved machineries from yeast to humans, namely
The ubiquitin–proteasome system (UPS) and;The autophagy–lysosomal system [179].

The UPS allows the specific elimination of ubiquitin-tagged protein by the 26S proteasome [180]. Ubiquitination is a three-step process involving three groups of enzymes (i.e., E1, E2 and E3) that catalyze the transfer of ubiquitin moieties or chains on lysine, serine, threonine or cysteine residues of target proteins (a modus operandi described in excellent reviews) [181,182]. The 26S proteasome is formed by 20S core complex and a 19S regulatory particle [183,184]. While the 20S core complex contains the three different catalytic activities that permit degradation, the 19S regulatory particle is specialized in the recognition and unfolding of ubiquitin-modified substrates [185]. Specifically, the 20S proteasome is a large complex comprised of two outer *α*-subunits rings (*α*1–7) embracing two central head-to-head oriented rings containing *β*-subunits (*β*1–7). The catalytic activity is conferred by three of the *β* subunits, namely *β*1, *β*2 and *β*5 which exhibit caspase-like, trypsin-like and chymotrypsin-like activity, respectively [186]. The 19S regulatory particle is made up of a base containing six ATPase subunits (Rpt1–6) which unfold the protein substrate and a lid carrying subunits (i.e., Rpn10, Rpn13) capable of binding ubiquitin-modified proteins [187]. Importantly, the *β*1, *β*2 and *β*5 standard subunits may be replaced by the so-called *β*1i, *β*2i and *β*5i inducible ones in newly synthetized complexes to promote a switch from constitutive proteasomes to immunoproteasomes [188,189]. It is appreciated that immunoproteasomes are more effective than their standard counterparts at degrading ubiquitin-modified proteins, making them particularly important in situations in which protein homeostasis is challenged such as during infection or oxidative stress [190,191,192,193,194,195]. Due to its unique position at the intersection of multiple pathways [196], the UPS is involved in the regulation of myriad of cellular processes, and as such any dysfunction of one of its components may carry serious consequences on cell functioning or viability.

#### 3.3.1. The UPS and its Dysfunctions

One major cause of proteasome dysfunction is aging [197]. Although not always consistent, investigations in animals and patients suffering from sporadic aging-related neurodegenerative diseases point to a decline ability of the UPS to eliminate undesirable proteins [198,199,200,201,202,203,204,205]. Other causes of UPS impairment are loss-of-function mutations in genes encoding proteasome subunits [35]. Surprisingly, as shown in Table 1, such genomic alterations lead to two clinical distinct phenotypes, namely autoinflammation (CANDLE/PRAAS) and neurodevelopmental disorders (NDD).

#### 3.3.2. Proteasome-Associated Autoinflammatory Syndromes (PRAAS)

From the mid-eighties, an increasing number of cases of rare autoinflammatory syndromes sharing similar clinical features such as lipodystrophy, skin lesions and recurrent fever have been reported. These conditions were referred to by a number of different names including Nakajo–Nishimura syndrome (NNS) [228,229,230], joint contractures, muscle atrophy and panniculitis-induced lipodystrophy (JMP) syndrome [230] and chronic atypical neutrophilic dermatitis with lipodystrophy and elevated temperatures (CANDLE) [231,232]. From 2010, it became evident that such syndromes were caused by loss-of-function mutations in genes encoding proteasome subunits and/or proteasome assembly factors and were subsequently renamed proteasome-associated autoinflammatory syndromes (PRAAS) [207,210,211,212,220,221,223,233] (Table 1). Two major hallmarks of PRAAS are the presence of a constitutive type I IFN gene signature in patients’ blood cells and a concomitant increased accumulation of ubiquitin-modified proteins in various cell types including fibroblasts. Recent studies have identified cells of the hematopoietic lineage as decisive mediators of type I IFN in PRAAS [208,234] and in view of increased eIF2α phosphorylation in B cells isolated from PRAAS subjects [221], a possible role of the UPR has been suggested in the induction of autoinflammation in these patients [35]. This notion was supported by in vitro studies showing that blocking the IRE1 arm of the UPR prevents the upregulation of IFN-stimulated genes (ISG) in response to proteasome inhibition in microglia cells [235]. Later studies however have shown that the IL-24/PKR axis of the ISR was the key contributor to type I IFN responses in ten unrelated PRAAS subjects [59,236]. These works unambiguously identify cytosolic IL-24 and PKR as a new DAMP/PRR pair involved in proteostasis surveillance. Whether other mechanisms contribute to sterile inflammation in PRAAS, notably in cells expressing no or low levels of IL-24, is currently not known.

#### 3.3.3. Neurodevelopmental Disorders (NDD) Caused by Proteasome Variants

Surprisingly, the phenotypic spectrum of proteasome loss-of-function mutations is broader than initially assumed. In 2017, the identification of pathogenic variants of the *PSMD12* gene in patients showing a predominant neuronal phenotype came as a surprise [224]. Subjects with *PSMD12* mutations presented with short stature, facial dysmorphism, intellectual disability, limb anomalies and sometimes absent of speech. Later, similar manifestations were reported in patients carrying variants of the *PSMB1* and *PSMC3* genes [206,227], thereby confirming the causal relationship between proteasome loss-of-function and NDD. One particularly intriguing observation from these studies is that, in contrast to PRAAS patients, subjects with NDD were devoid of clinical signs of autoinflammation. However, recent investigation revealed that leukocytes isolated from NDD patients with *PSMD12* haploinsufficiency do exhibit a typical type I IFN gene signature [68]. The reason why the upregulation of ISG in these cells is not associated with the development of clinical autoinflammation as observed in PRAAS patients is obscure, but this phenomenon has been described in other interferonopathies [237,238,239,240]. Interestingly, it was shown that *PSMD12* haploinsufficiency was accompanied by a profound remodeling of mTORC1 and its downstream autophagy pathway. Specifically, the elimination of mitochondria by mitophagy in *PSMD12*-deficient cells was induced [68].

#### 3.3.4. Disorders due to Deficient DUB and/or E3 Ubiquitin Ligases

A number of autoinflammatory syndromes have been described to be caused by genomic alterations in genes coding for DUB and E3 ubiquitin ligases. These include notably loss- or gain-of-function mutations in the *TNFAIP3*, *USP18* and *OTULIN*, *RNF31*, *RBCK1* genes [241,242,243,244,245,246,247]. However, in contrast to their proteasome counterparts, pathogenic variants from DUB and/or E3 ubiquitin ligases exert their proinflammatory effects mostly by interfering with specific PRR signaling cascades rather than by destabilizing the whole-cell proteome [248].

#### 3.3.5. The Autophagy-Lysosomal System and Its Defects

The 2016 Nobel Prize in Physiology or Medicine went to Yoshinori Ohsumi for his discoveries of the “mechanisms of autophagy” [249,250]. Together with the UPS, autophagy is the main contributor to protein breakdown in the cell. Unlike the UPS, which is constitutive active, autophagy requires signals such as starvation, growth, etc. It relies on the formation of autophagosomes capable of sequestering intracellular material prior to delivery to lysosomes for subsequent degradation. Over the past few years, various types of autophagy have been described including micro-autophagy and macro-autophagy [251], the latter enabling the elimination of protein aggregates [252], a process frequently referred to as “aggrephagy” [253]. In this selective form of autophagy, ubiquitin-modified protein aggregates are recognized by the autophagy receptors p62, NBR1 and OPTN which themselves interact with lipidated LC3II on the inner phagophore membrane [254], thereby targeting them to autophagosomes. As such, protein homeostasis critically depends on functional autophagy under stress conditions and any and loss of autophagy has been early associated with the aggregation of abnormal proteins and neurodegeneration [255].

Another major cause of autophagy dysfunction and a fortiori protein homeostasis perturbations are loss-of-function mutations in genes encoding any one of the many components involved in this process. A particularly vulnerable gene in this regard is *SQSTM1* encoding the autophagy receptor p62 and whose pathogenic variants are associated with the onset of various metabolic, myopathic and skeletal disorders as well as neurodegenerative disease including ALS. Strikingly, syndromes caused by SQSTM1 disruption are clearly associated with signs of autoinflammation [256,257,258]. However, given that p62 is also directly involved in the regulation of NF-κB signaling [259,260], the precise contribution of impaired aggrephagy to inflammation in these diseases is difficult to assess. Other ALS genes potentially causing imbalanced protein homeostasis include ATG5 and ATG7 [261,262]. Although clearly associated with protein aggregates, no inflammation was observed in these patients.

## 4. Conclusions and Future Directions

Altogether, these studies revealed that eukaryotic cells are equipped with various sensitive systems capable of sensing quantitative aberrations of their intra- and extracellular proteomes. Strikingly, these sensors are more or less directly connected to innate signaling pathways, and the continuous sampling of proteostasis perturbations consistently results in autoinflammation. The reason why the cell mostly responds to impaired proteostasis by the production of proinflammatory mediators is however not fully understood. One possible explanation is that inflammation might be part of autocrine program primarily aimed to help restoring proteostasis. This hypothesis is particularly based on the fact that many of the target genes (i.e., ISG) of type I IFN encode products that act on the proteostasis network at various levels. For instance, type I IFN limits the proteostatic burden by supporting a translation arrest via the induction of PKR. Conversely, by inducing immunoproteasome subunits and proteasome activators [263], type I IFN accelerates protein breakdown and as such help the cells to cope with proteotoxic stress. Given that inflammation has proliferation-promotive effects [264], it might also well be that the initiation of innate responses under these circumstances is intended to preserve integrity by generating new cellular space for the accumulating protein aggregates. Herein, when limited in time, sterile inflammation might represent a beneficial reaction rebalancing dysregulated proteostasis. However, the onset of sterile autoinflammation or neuroinflammation in patients with impaired proteostasis might signify a point of no return from which protein homeostasis cannot be rescued any longer. In this regard, a better comprehension of the mechanisms driving these processes is needed.

## Figures and Tables

**Figure 1 cells-11-01422-f001:**
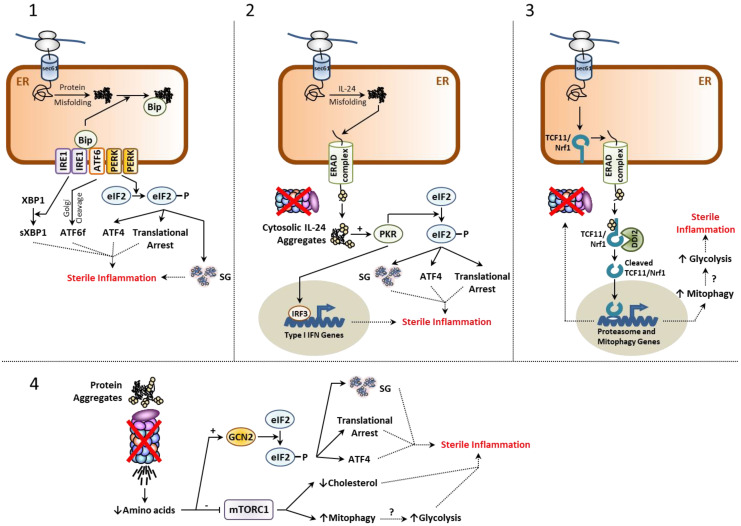
Most inflammatory pathways engaged following proteostasis disruption originate from the ER. Proteostasis perturbations are typically caused by proteasome dysfunction and/or excessive protein misfolding, as indicated. (**1**) Sustained misfolding of secretory proteins results in their accumulation within the ER lumen and generates ER stress, which is sensed by the IRE1, ATF6 and PERK receptors upon dissociation of the chaperone protein Bip. This, in turn, initiates the UPR that comprises the formation of sXBP1, ATF6f as well as a translational arrest, the activation of ATF4 and the formation of SGs following phosphorylation of eIF2*α* by PERK. Persistent activation of all UPR branches have been reported to trigger sterile inflammation by various mechanisms (for more details, see main text). (**2**) Misfolding of IL-24 typically results in its subsequent retro-translocation back to the cytosol by the ERAD machinery. In case of proteasome dysfunction, IL-24 misfolded species accumulate in the cytosol and activate PKR to trigger a type I IFN response and inflammation via eIF2α phosphorylation. (**3**) Proteome imbalance is also sensed by the short-lived ER-membrane resident protein TCF11/Nrf1 whose turnover rate decreases with increased proteasome dysfunction. This results in its proteolytic cleavage by DDI2, thereby giving rise to a transcription factor inducing the expression of proteasome genes to restore protein homeostasis and mitophagy genes that may exacerbate inflammation through persistent glycolysis, as indicated. (**4**) Proteasome defects also reduce the intracellular pool of free amino acids and result in mTORC1 downregulation. This, in turn, promotes sterile inflammation by increasing mitophagy and blocking lipid synthesis (for more details, see the main text). Amino acid depletion in the cell also activates GCN2 of the integrated stress response (ISR) which facilitates inflammation following eIF2α phosphorylation.

**Figure 2 cells-11-01422-f002:**
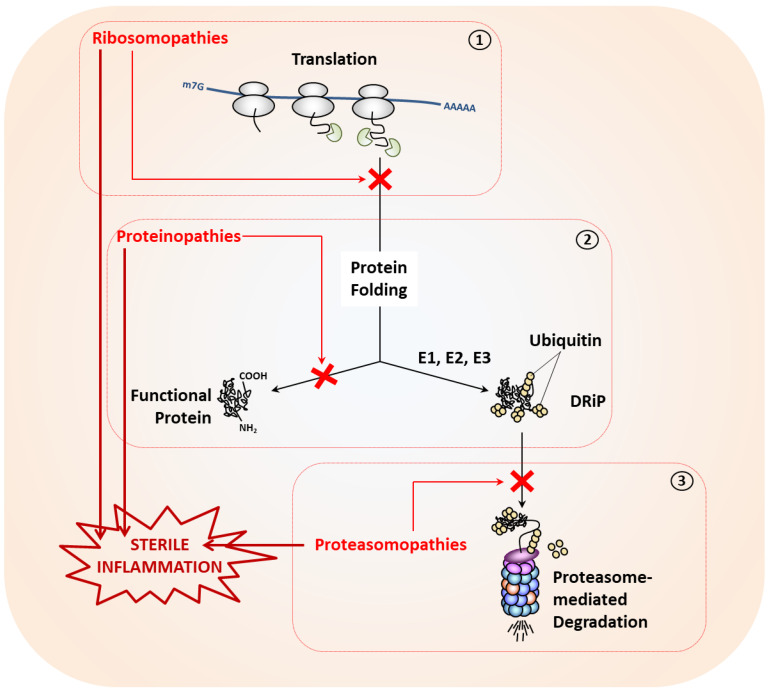
Proteostasis perturbations may arise at any stage of the protein life cycle and typically result in the initiation of sterile inflammation. Depicted are three pathological situations causing impaired proteostasis in which translation (**1**), protein folding (**2**) and proteasome-mediated protein degradation of defective ribosomal products (DRiPs) following their polyubiquitination by E1, E2 and E3 enzymes of the ubiquitin-conjugation pathway (**3**) are affected.

**Figure 3 cells-11-01422-f003:**
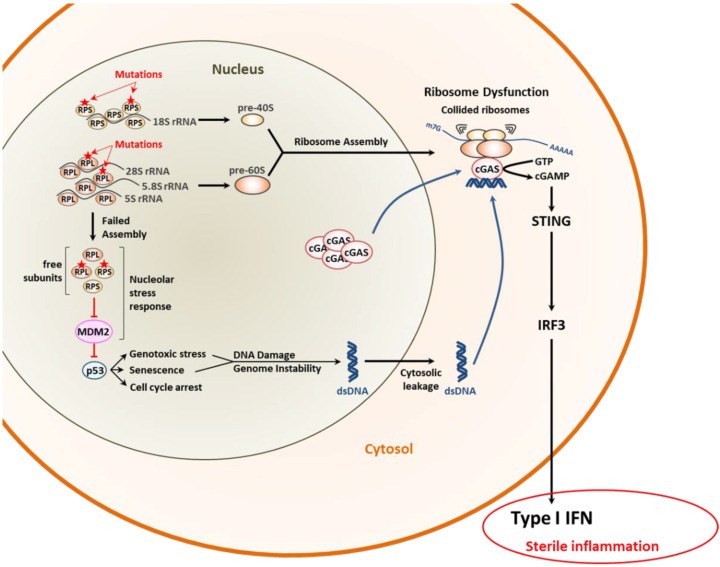
Molecular pathogenesis of ribosomopathies and their potential inflammatory consequences. The failure of mutant ribosomal proteins of the small 40S (RPS) and/or large 60S (RPL) ribosomal subunits to assemble into ribosomes induces the nucleolar stress response leading to p53 stabilization via persistent inactivation of the MDM2 E3 ubiquitin ligase. This, in turn, results in genome instability and subsequent leakage of double-stranded (ds)DNA into the cytosol. Cytoplasmic dsDNA is then sensed by the cyclic GMP-AMP synthase (cGAS) which promotes a type I IFN response via the STING/IRF3 signaling pathway, as indicated. The recruitment of cGAS in the cytosol is itself facilitated by defective and collided ribosomes.

**Figure 4 cells-11-01422-f004:**
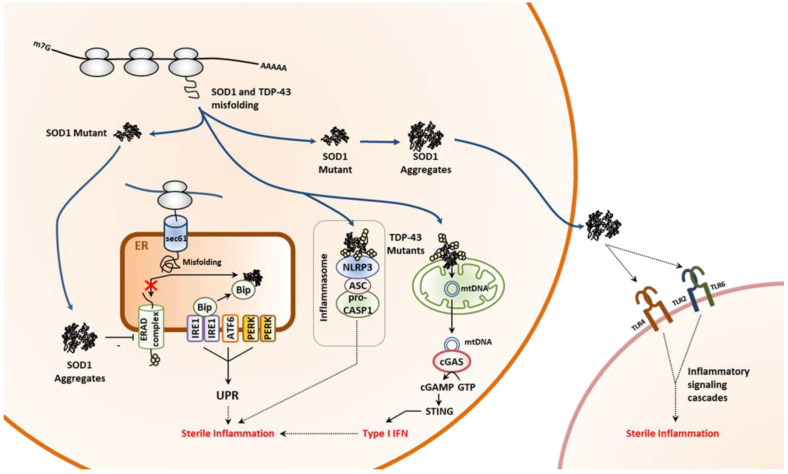
Neuroinflammation caused by impaired proteostasis in ALS involves multiple pathways. Pathogenic variants of superoxide dismutase 1 (SOD1) facilitate the formation of intracellular protein aggregates which themselves promote sterile inflammation by causing persistent activation of the unfolded protein response (UPR) via inhibition of the ER-associated degradation machinery (ERAD), as indicated. In addition, extracellular SOD1 protein aggregates exert their pro-inflammatory effects by binding to phagocyte pattern recognition receptors such as Toll-like receptors (TLR)-2, 4 and 6. Pathogenic mutations within the TAR DNA-binding protein 43 (TDP-43) results in protein aggregation which activates the inflammasome and triggers the release of mitochondrial (mt) DNA. The latter, in turn, initiates a type I IFN response following sensing by the cytosolic cyclic GMP-AMP (cGAS) synthase and subsequent STING activation, as indicated. NLRP3: NOD-like receptor pyrin domain-containing-3; ASC: apoptosis associated speck-like protein containing a CARD; pro-CASP1: pro-caspase 1.

**Table 1 cells-11-01422-t001:** Summary of the genetic characteristics and associated clinical phenotypes of the proteasome variants identified so far.

Proteasome	Gene	Variant	Genetic Model	Origin	Phenotype	Reference
20S Complex	*PSMB1*	p.Y103H	Homozygous, monogenic	Recessive inheritance	NDD	[206]
	*PSMB4*	5′ UTR: c.–9G > A	Compound heterozygous, monogenic	Recessive inheritance	PRAAS	[207]
		p.D212_V214del				
	*PSMB4*	p.L78Wfs * 31	Compound heterozygous, monogenic	Recessive inheritance	PRAAS	[208]
		c.494 + 17A > G				
	*PSMB4/* *PSMB8*	p.Y222 *	Double heterozygous, digenic	Recessive inheritance	PRAAS	[207]
		p.K105Q				
	*PSMB8*	p.G179V	Homozygous, monogenic	Recessive inheritance	PRAAS	[209]
	*PSMB8*	p.G201V	Homozygous, monogenic	Recessive inheritance	PRAAS	[210]
	*PSMB8*	p.C135 *	Homozygous, monogenic	Recessive inheritance	PRAAS	[211]
	*PSMB8*	p.T75M	Homozygous, monogenic	Recessive inheritance	PRAAS	[212]
	*PSMB8*	p.R125C	Compound heterozygous, monogenic	Recessive inheritance	PRAAS	[213]
		p.D119N				
	*PSMB8*	p.Q55 *	Compound heterozygous, monogenic	Recessive inheritance	PRAAS	[214]
		p.S118P				
	*PSMB8*	p.A92V	Compound heterozygous, monogenic	Recessive inheritance	PRAAS	[215]
		p.K105Q				
	*PSMB8*	p.A92T	Homozygous, monogenic	Recessive inheritance	PRAAS	[216]
	*PSMB8*	-	Homozygous, monogenic	Recessive inheritance	PRAAS	[217]
	*PSMB8/* *PSMA3*	p.T75M	Double heterozygous, digenic	Recessive inheritance	PRAAS	[207]
		p.H111Ffs * 10				
	*PSMB8/* *PSMA3*	p.T75M	Double heterozygous, digenic	Recessive inheritance	PRAAS	[207]
		p.R233del				
	*PSMB9/* *PSMB4*	p.G165D	Double heterozygous, digenic	Recessive inheritance	PRAAS	[207]
		p.P16Sfs * 45				
	*PSMB9*	p.G156D	Heterozygous, monogenic	de novo, dominant	PRAAS	[218] [219]
	*PSMB10*	p.F14S	Homozygous, monogenic	Recessive inheritance	PRAAS	[220]
Assembly Factors	*POMP*	p.E115Dfs * 20	Heterozygous, monogenic	de novo, dominant	PRAAS	[207]
	*POMP*	p.F114Lfs * 18	Heterozygous, monogenic	de novo, dominant	PRAAS	[221]
	*POMP*	p.I112Wfs * 3	Heterozygous, monogenic	de novo, dominant	PRAAS	[221]
	*POMP*	p.D109Efs * 2	Heterozygous, monogenic	de novo, dominant	PRAAS	[222]
	*PSMG4*	p.Y223Sfs * 2	Compound heterozygous, monogenic	Recessive inheritance	PRAAS	[223]
		p.N225K				
19S Complex	*PSMD12*	p.R123 *	Heterozygous, monogenic	de novo, dominant	NDD	[224]
	*PSMD12*	p.L425 *	Heterozygous, monogenic	de novo, dominant	NDD	
	*PSMD12*	p.R201 *	Heterozygous, monogenic	de novo, dominant	NDD	
	*PSMD12*	c.909−2A > G	Heterozygous, monogenic	de novo, dominant	NDD	
	*PSMD12*	Deletion	Heterozygous, monogenic	de novo, dominant	NDD	
	*PSMD12*	p.R201 *	Heterozygous, monogenic	de novo, dominant	NDD	[225]
	*PSMD12*	p.R182 *	Heterozygous, monogenic	de novo, dominant	NDD	[68]
	*PSMD12*	p.R357fs * 3	Heterozygous, monogenic	de novo, dominant	NDD	
	*PSMD12*	p.T146Kfs * 3	Heterozygous, monogenic	de novo, dominant	NDD	
	*PSMD12*	p.E313 *	Heterozygous, monogenic	de novo, dominant	NDD	
	*PSMD12*	p.Q170Gfs * 40	Heterozygous, monogenic	de novo, dominant	NDD	
	*PSMD12*	p.L149 *	Heterozygous, monogenic	de novo, dominant	NDD	
	*PSMD12*	p.Q106 *	Heterozygous, monogenic	de novo, dominant	NDD	
	*PSMD12*	p.Q345 *	Heterozygous, monogenic	de novo, dominant	NDD	
	*PSMD12*	c.1083 + 1G > A	Heterozygous, monogenic	de novo, dominant	NDD	
	*PSMD12*	p.Q416 *	Heterozygous, monogenic	de novo, dominant	NDD	
	*PSMD12*	p.S176Qfs * 15	Heterozygous, monogenic	de novo, dominant	NDD	
	*PSMD12*	c.1162−1G > A	Heterozygous, monogenic	de novo, dominant	NDD	
	*PSMD12*	p.S434Hfs * 2	Heterozygous, monogenic	de novo, dominant	NDD	
	*PSMD12*	c.795 + 1G > A	Heterozygous, monogenic	de novo, dominant	NDD	
	*PSMD12*	p.L50Gfs * 26	Heterozygous, monogenic	de novo, dominant	NDD	
	*PSMD12*	p.T146Kfs * 3	Heterozygous, monogenic	de novo, dominant	NDD	
	*PSMD12*	p.R182 *	Heterozygous, monogenic	de novo, dominant	NDD	
	*PSMD12*	p.L354Efs * 6	Heterozygous, monogenic	de novo, dominant	NDD	
	*PSMD12*	p.Y302 *	Heterozygous, monogenic	de novo, dominant	NDD	
	*PSMD12*	p.R289 *	Heterozygous, monogenic	de novo, dominant	NDD	[226]
	*PSMC3*	p.S376Rfs15 *	Homozygous, monogenic	Recessive inheritance	NDD	[227]

* Asterisks indicate termination codons.

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
