# Peer review of "Proteostasis Perturbations and Their Roles in Causing Sterile Inflammation and Autoinflammatory Diseases"

_cells, 2022, doi:10.3390/cells11091422_

Round 1
Reviewer 1 Report
This review summarises the current knowledge on the role of proteasome dysfunction and the associated pathologies in particular inflammation processes.
The review is well written, but I would suggest to focus on signalling pathways/cellular circuits on the regulation of the proteasome. In its current form, the review is too long, too broad and hard to digest.
I also recommend to refrain from bold statements such as in the abstract "Cell integrity and proper functioning critically depend on the maintenance of a constant intracellular protein pool" What is the evidence for that?
same for the next sentence: "In eukaryotic cells, this equilibrium is determined by an equal balance between protein synthesis and degradation,..." Again what is the evidence that there is an equilibrium?
Other than that, this review is very informative. My only 2 recommendations are:
- Focus on 1-2 aspects - do not cover the entire knowledge of proteasome signalling, dysfunction, protein synthesis regulation and imbalance and all disease and pathological processes.
- refine the figures. There are too complex and could be improved in general with resources from Bioicon or BioRender etc.
Author Response
Reviewer #1
This review summarises the current knowledge on the role of proteasome dysfunction and the associated pathologies in particular inflammation processes. The review is well written, but I would suggest to focus on signalling pathways/cellular circuits on the regulation of the proteasome. In its current form, the review is too long, too broad and hard to digest.
>> We agree wit the referee that the original version of our manuscript was probably too long (7.850 words) and sometimes hard to follow. We followed his/her excellent suggestion and cut this paper down to 6.425 words by removing parts of sections 3.2.1 and 3.2.2 dedicated to the off-topic -and through- description of the pathogenesis of ankylosing spondylitis (AS), cystic fibrosis (CF), alpha-1-antitrypsin deficiency (AATD) Alzheimer disease (AD), Parkinson disease (PD), Huntington disease (HD) (lines 349-353 and 376-382).
I also recommend to refrain from bold statements such as in the abstract "Cell integrity and proper functioning critically depend on the maintenance of a constant intracellular protein pool" What is the evidence for that?
>> We thank the reviewer for this comment and changed this sentence in the revised version of the manuscript accordingly (lines 9-10).
same for the next sentence: "In eukaryotic cells, this equilibrium is determined by an equal balance between protein synthesis and degradation,..." Again what is the evidence that there is an equilibrium?
>> This has been changed (lines 9-10).
Other than that, this review is very informative. My only 2 recommendations are:
Focus on 1-2 aspects - do not cover the entire knowledge of proteasome signalling, dysfunction, protein synthesis regulation and imbalance and all disease and pathological processes.
>> The referee raises here a valid point. As mentioned above, we have shortened our manuscript by more than 1.000 words. The revised version of this paper is now devoid of unnecessary statements in sections 3.2.1 and 3.2.2 which exclusively focus on the mechanisms by which sterile inflammation is triggered in neurodegenerative and non-neurodegenerative proteinopathies (lines 349-353 and 376-382).
refine the figures. There are too complex and could be improved in general with resources from Bioicon or BioRender etc.
>> We have substantially simplified all figures, as suggested by the reviewer.

Reviewer 2 Report
This is an interesting, original and timely review on a medically highly relevant topic. While the importance of protein homeostasis in the context of neurodegeneration is very frequently covered by review articles, its link to sterile inflammation and autoinflammatory diseases is clearly underexplored and under-reviewed.
After an introduction into inflammation in general and DAMP-induced sterile inflammation in particular, the authors describe in the first part of the review signaling pathways that link proteostasis to sterile inflammation, including the UPR, the ISR, mTORC1 signaling, the NFE2L1 pathway and some more. In the larger second part of the review, they discuss in detail how different types of proteostasis perturbations are linked to autoinflammatory syndromes, with a focus on neurodegenerative diseases and impaired proteasome function.
Overall, the review follows a clear concept and should be of interest for a broad readership. However, there are a few issues that should be addressed by the authors.
Major points:
- Not all of the figures adequately support the text. Figure 1 is way too complicated, does not follow a consistent style (e.g. for compartments and proteins; "ERAD" seems to be just another protein sitting in the ER membrane), and is in large parts redundant with Figure 4. It would probably be better to subdivide this figure in several sub-panels depicting certain aspects of what is shown. Moreover, some details of Figs. 1/4 are oversimplified/incorrect: ATF6 activation involves transport to and cleavage at the Golgi; the UPR sensors are not directly activated by aggregates (but by loss of BiP binding); CFTR mutants do not form luminal aggregates (lines 348/349).
Figure 2 contains little information relative to its size. The polysome representation in box 1 does not match the perturbations described in the text and Figure 3, i.e. nucleolar assembly defects leading to DAMP formation and ribosome collisions.
- The focus is quite narrow at some points. In section 3.2 (protein misfolding), the central role of molecular chaperones is not even mentioned. In section 3.3 (impaired protein degradation), the focus is entirely on the 26S proteasome, even though a number of E3s and DUBs are known to be involved in perturbed proteostasis and/or autoinflammation.
- Sections 3.2.1 and 3.2.2 contain a long list of disorders linked to perturbed proteostasis, with more or less clear links to sterile inflammation. Perhaps it would be instructive to discuss 1-2 examples where these links are better understood in more detail.
Further points:
- Line 91: When introducing the UPR, the groundbreaking work of David Ron and Peter Walter should be cited.
- Lines 186/187: Gives the impression that SGs are an exclusive feature of NDDs. However, they form in all cells under certain stress conditions.
- Lines 425ff: Gives the impression that PARK1 is the only gene mutated in PD. What about all the other PARKs, e.g. Parkin/PARK2?
- Section 3.2.2 adds relatively little to the review, because the links to sterile inflammation appear to be rather weak. Remove?
- There are inconsistencies in the notation of gene names (should all be italic) and abbreviations (e.g. AT vs AAT). Also, some abbreviations are not defined at first mention (AD), and some terms are explained twice (mitophagy).
- There are a number of typos and missing or superfluous words.
Author Response
Reviewer 2
This is an interesting, original and timely review on a medically highly relevant topic. While the importance of protein homeostasis in the context of neurodegeneration is very frequently covered by review articles, its link to sterile inflammation and autoinflammatory diseases is clearly underexplored and under-reviewed.
After an introduction into inflammation in general and DAMP-induced sterile inflammation in particular, the authors describe in the first part of the review signaling pathways that link proteostasis to sterile inflammation, including the UPR, the ISR, mTORC1 signaling, the NFE2L1 pathway and some more. In the larger second part of the review, they discuss in detail how different types of proteostasis perturbations are linked to autoinflammatory syndromes, with a focus on neurodegenerative diseases and impaired proteasome function.
Overall, the review follows a clear concept and should be of interest for a broad readership. However, there are a few issues that should be addressed by the authors.
Major points:
Not all of the figures adequately support the text. Figure 1 is way too complicated, does not follow a consistent style (e.g. for compartments and proteins; "ERAD" seems to be just another protein sitting in the ER membrane), and is in large parts redundant with Figure 4. It would probably be better to subdivide this figure in several sub-panels depicting certain aspects of what is shown. Moreover, some details of Figs. 1/4 are oversimplified/incorrect: ATF6 activation involves transport to and cleavage at the Golgi; the UPR sensors are not directly activated by aggregates (but by loss of BiP binding); CFTR mutants do not form luminal aggregates (lines 348/349).
>> We took the referee’s remark into consideration and subdivided figure 1 into 4 distinct subpanels in the revised version of this manuscript (page 3). We also corrected the depiction of ERAD, ATF6 according to referee’s recommendations (page3).
Figure 2 contains little information relative to its size. The polysome representation in box 1 does not match the perturbations described in the text and Figure 3, i.e. nucleolar assembly defects leading to DAMP formation and ribosome collisions.
>> We addressed the reviewer’s concern regarding the size of figure. 2 which is now reduced in the revised version of this manuscript (page 6). Nevertheless, we decided not to change the polysome depicted in this figure, as it represents normal protein translation giving rise to functional proteins, as indicated in panel 2. We believe that adding a defective polysome into this figure would confuse the reader at this point.
The focus is quite narrow at some points. In section 3.2 (protein misfolding), the central role of molecular chaperones is not even mentioned. In section 3.3 (impaired protein degradation), the focus is entirely on the 26S proteasome, even though a number of E3s and DUBs are known to be involved in perturbed proteostasis and/or autoinflammation.
>> The referee is correct. To tackle this point, we have added a new paragraph on DUB and/or E3 ligases deficiencies in section 3.3.1 (lines 506-512). We, however, emphasized that, unlike proteasome loss-of-function mutations, genomic alterations of DUB or E3 ubiquitin ligases trigger autoinflammation by specifically interfering with signaling cascades engaged by pattern recognition receptors (PRR) rather than by destabilizing the whole-cell proteome.
Sections 3.2.1 and 3.2.2 contain a long list of disorders linked to perturbed proteostasis, with more or less clear links to sterile inflammation. Perhaps it would be instructive to discuss 1-2 examples where these links are better understood in more detail.
>> We agree with reviewer that we failed to focus our analysis on a single key message when discussing the various disorders caused by perturbed proteostasis. To meet the referee’s requirements, we have removed substantial parts of sections 3.2.1 and 3.2.2 and focused our investigations on the mechanisms by which sterile inflammation is triggered in these diseases (lines 349-353 and 376-382).
Further points:
Line 91: When introducing the UPR, the groundbreaking work of David Ron and Peter Walter should be cited.
>> We thank the reviewer for this excellent suggestion. We have now cited two review articles of David Ron and Peter Walter on this topic in the revised version of this manuscript (line 91).
Lines 186/187: Gives the impression that SGs are an exclusive feature of NDDs. However, they form in all cells under certain stress conditions.
>> The referee raised a valid point here. We took his/he remark into consideration in the revised version of the manuscript (lines 194-197).
Lines 425ff: Gives the impression that PARK1 is the only gene mutated in PD. What about all the other PARKs, e.g. Parkin/PARK2?
>> To address the issue of manuscript length raised by both reviewers, this part of the manuscript has been removed in the revised version of our manuscript (lines 349-353 and 376-382).
Section 3.2.2 adds relatively little to the review, because the links to sterile inflammation appear to be rather weak. Remove?
>> This has been changed (see above).
There are inconsistencies in the notation of gene names (should all be italic) and abbreviations (e.g. AT vs AAT). Also, some abbreviations are not defined at first mention (AD), and some terms are explained twice (mitophagy).
>> We thank the reviewer for pointing this out. We have carefully checked the revised version of our manuscript for gene annotation and abbreviations.
There are a number of typos and missing or superfluous words.
>> We are grateful to the referee for pointing this out. The revised version of our manuscript has been triple-checked for typos.
